# Platelet-Rich Plasma (PRP) and Adipose-Derived Stem Cell (ADSC) Therapy in the Treatment of Genital Lichen Sclerosus: A Comprehensive Review

**DOI:** 10.3390/ijms242216107

**Published:** 2023-11-09

**Authors:** Alessia Paganelli, Luca Contu, Alessandra Condorelli, Elena Ficarelli, Alfonso Motolese, Roberto Paganelli, Alberico Motolese

**Affiliations:** 1Dermatology Unit, Arcispedale Santa Maria Nuova, Istituto di Ricovero e Cura a Carattere Scientifico-Azienda Unità Sanitaria Locale di Reggio Emilia, 42123 Reggio Emilia, Italy; alessia.paganelli@ausl.re.it (A.P.); luca.contu@ausl.re.it (L.C.); alessandra.condorelli@ausl.re.it (A.C.); elena.ficarelli@ausl.re.it (E.F.); alberico.motolese@ausl.re.it (A.M.); 2Department of Clinical and Experimental Medicine, Section of Dermatology, Università degli Studi di Messina, 98122 Messina, Italy; alfonsomotolese93@gmail.com; 3Internal Medicine, UniCamillus International Medical University in Rome, 00131 Rome, Italy

**Keywords:** lichen sclerosus, PRP, ADSC, mesenchymal stromal cells, regenerative medicine, vulvar atrophy

## Abstract

Lichen sclerosus (LS) is a chronic inflammatory dermatosis mostly localized in the genital area, characterized by vulvar alterations that can severely impact a patient’s quality of life. Current treatment modalities often provide incomplete relief, and there is a need for innovative approaches to manage this condition effectively. Platelet-rich plasma (PRP) and adipose-derived stem cells (ADSCs) have emerged as potential regenerative therapies for LS, offering promising results in clinical practice. This comprehensive review explores the utilization of PRP and ADSC therapy in the treatment of genital LS, highlighting their mechanisms of action, safety profiles, and clinical outcomes. PRP is a blood product enriched in growth factors and cytokines, which promotes tissue regeneration, angiogenesis, and immune modulation. ADSC regenerative potential relies not only in their plasticity but also in the secretion of trophic factors, and modulation of the local immune response. Numerous studies have reported the safety of PRP and ADSC therapy for genital LS. Adverse events are minimal and typically involve mild, self-limiting symptoms, such as transient pain and swelling at the injection site. Long-term safety data are encouraging, with no significant concerns identified in the literature. PRP and ADSC therapy have demonstrated significant improvements in LS-related symptoms, including itching, burning, dyspareunia, and sexual function. Additionally, these therapies enable many patients to discontinue the routine use of topical corticosteroids. Several studies have explored the efficacy of combining PRP and ADSC therapy for LS. In combination, PRP and ADSCs seem to offer a synergistic approach to address the complex pathophysiology of LS, particularly in the early stages. The use of PRP and ADSC therapy for genital lichen sclerosus represents a promising and safe treatment modality. These regenerative approaches have shown significant improvements in LS-related symptoms, tissue trophism, and histological features. Combination therapy, which harnesses the synergistic effects of PRP and ADSCs, is emerging as a preferred option, especially in early-stage LS cases. Further research, including randomized controlled trials and long-term follow-up, is warranted to elucidate the full potential and mechanisms of PRP and ADSC therapy in the management of genital LS. These regenerative approaches hold great promise in enhancing the quality of life of individuals suffering from this challenging condition.

## 1. Introduction

Lichen sclerosus (LS) is a relatively rare and often misdiagnosed dermatological condition, that primarily affects the skin in the genital and perianal region [1]. While it is not a life-threatening condition, its impact on patient quality of life and psychological well-being should not be underestimated [2]. At its core, lichen sclerosus is a chronic inflammatory skin disorder that manifests as thin, white patches on the skin’s surface. These patches, often described as looking like “cigarette paper” or “parchment”, are characterized by their smooth, shiny appearance and can appear anywhere on the body [3]. However, they most commonly affect the genital and perianal regions in both sexes, leading to considerable discomfort and distress. Although less common, lichen sclerosus can also involve extragenital areas, such as the breasts, upper body, and thighs [4]. These extragenital manifestations can sometimes mimic other skin conditions, making it crucial for medical professionals to be well-versed in recognizing and differentiating lichen sclerosus. One of the hallmark symptoms of lichen sclerosus is pruritus [5]. Itching can be severe and debilitating, significantly impacting patient’s daily life [6]. Additionally, scratching can worsen skin inflammation and may result in further complications, such as fissures, bleeding, and even pain during sexual intercourse [7]. In addition to the physical symptoms, the emotional toll of LS should not be underestimated. Living with a chronic, often misunderstood condition can lead to anxiety, depression, and reduced self-esteem [8,9]. Diagnosing lichen sclerosus can be challenging, and misdiagnoses are not uncommon. Even expert dermatologists, in fact, may need to perform a biopsy to confirm this condition.

While the exact cause of lichen sclerosus remains unclear, several factors seem to contribute to its development. There is evidence to suggest that autoimmune factors may play a role, as the condition is associated with other autoimmune diseases such as thyroid disorders and vitiligo [10,11]. Hormonal imbalances are also believed to be a contributing factor, which could at least partially explain the higher prevalence of LS in postmenopausal women [12]. Genetics seem to also possibly play a role. However, further research is needed to fully understand the complex interplay of these factors in LS pathogenesis.

Topical corticosteroid creams or ointments are commonly prescribed to reduce inflammation and alleviate itching [13]. In some cases, other immunomodulatory medications may also be recommended to help manage the condition, such as topical calcineurin inhibitors or systemic immunosuppressants [14]. Despite optimal treatment, LS often poses significant challenges in terms of symptom control and management, since conventional therapies lead to limited success in a non-negligible proportion of cases. In fact, complete remission of LS-related signs and symptoms remains an unreachable goal for most of the patients. Since there is currently no availability of a definitive cure for LS, new strategies have recently been investigated; currently, LS treatment represents an expanding field of research. More specifically, platelet-rich plasma (PRP) therapy has emerged as a promising and innovative approach in the management of various dermatological conditions, and its potential benefits in treating LS are gaining increasing attention [15]. PRP therapy has generated interest for its regenerative properties and potential to alleviate the distressing symptoms associated with LS [15]. Nevertheless, adipose-tissue derived stromal cells (ADSCs) have also emerged as a novel therapeutic avenue, offering the potential not only to address the root causes of LS but also to promote tissue regeneration [16]. ADSC-based therapy is a cutting-edge and rapidly evolving field in regenerative medicine, which holds immense promise for individuals suffering from LS.

The aim of the present review is to provide an overview of current knowledge in the setting of LS specifically focusing on PRP and ADSCs, as these represent promising therapeutic tools in this setting. Table 1 briefly summarizes the retrieved casuistries after systematic revision of available publications on these topics (for further details on the search method, see Appendix B and Appendix A; for more details on the published studies, see Appendix A).

## 2. Platelet-Rich Plasma (PRP) Therapy

PRP is a hemoderivative rich in growth factors and cytokines, and is hypothesized to stimulate tissue regeneration and repair, making it a potential candidate for mitigating the symptoms and structural changes associated with LS [15,17].

Autologous and homologous PRP are two distinct approaches to harnessing the therapeutic potential of platelets in regenerative medicine. These techniques differ primarily in the source of the PRP and their applications. A notable advantage of autologous PRP therapy is its safety profile, as it utilizes the patient’s own blood components, minimizing the risk of adverse reactions or immune responses [18]. This type of PRP is already commonly used not only in dermatology but also in orthopedics and sports medicine with the aim of accelerating healing, reducing pain, and promoting tissue repair [19]. In contrast, homologous PRP is obtained from donor’s blood. This approach is less common and typically involves a careful screening process to ensure donor safety and compatibility. The use of homologous PRP is currently limited to specialized research settings but has been demonstrated to offer some advantages in terms of efficacy when compared to autologous PRP [20]. However, evidence on homologous PRP is currently limited to other research contexts since no clinical studies have been published so far on the use of homologous PRP in the setting of LS.

As previously mentioned, the therapeutic potential of PRP (either autologous or homologous) is closely related to their content rich in platelets [21,22]. Platelets release growth factors that stimulate cell proliferation and tissue regeneration, making PRP an attractive option for a variety of medical conditions (see Figure 1) [23].

With regards to LS, early investigations have reported encouraging results, particularly in terms of symptom relief. PRP therapy has shown the potential to reduce itching, pain, and discomfort, which are hallmark symptoms of lichen sclerosus [16]. Furthermore, PRP’s regenerative capacity appears to have a positive impact on the structural changes associated with LS [24]. Preliminary studies suggest that PRP may promote the restoration of normal tissue architecture, potentially reducing the risk of scarring and fibrosis [25]. This is a significant consideration, as fibrotic changes can lead to structural alterations, further complicating the management of lichen sclerosus. Moreover, PRP’s anti-inflammatory properties may contribute to alleviating the chronic inflammation characteristic of the condition. An Australian study, for example, followed up for two years with 28 patients with confirmed vulvar LS were treated with autologous PRP [26]. All the patients were non-responders to topical steroid treatment. The results showed that almost all patients experienced clinical improvement in the size of their LS lesions. In eight cases, the lesions completely disappeared, and 23 out of the 28 patients eliminated the need for further steroid therapy, with complete remission of LS-related symptoms in 15 cases and significant improvement in the rest of the study population. Recently, symptomatic relief after platelet-rich plasma infiltration in vulvar LS was also described [27] by a Spanish group. Medina Garrido and collaborators performed a pilot study aimed at assessing the impact of PRP injections on symptom relief in postmenopausal women with vulvar LS who had not responded well to topical corticosteroid treatment. Three PRP infiltrations were administered to 28 patients, and changes in symptom scores were monitored using the clinical scoring system for vulvar lichen sclerosus (CSS) over the course of a year. The results revealed that women in the study experienced statistically significant improvements in self-assessed LS symptoms, and these improvements were sustained throughout the year of monitoring.

Several authors also focused on histological changes induced by PRP. Goldstein and collaborators, in fact, performed a randomized double-blind placebo-controlled trial to evaluate the safety and the efficacy of PRP in vulvar LS and chose the grade of inflammation in post-treatment biopsy as a primary endpoint [28,29]. Despite the study succeeding in demonstrating higher effectiveness of PRP compared to saline injection, histological improvement in terms of inflammatory infiltrates was demonstrated only in 5 out of 19 treated women. The group of Tedesco et al. followed up 31 patients (13 males, 18 females) for 12 months after a 3-session treatment with autologous PRP injected into affected areas [30]. All the cases of PRP were confirmed through biopsy. Possible comorbidities included psoriasis, esophagitis, lichen planus, myasthenia gravis, Hashimoto’s disease, and atrophic gastritis. After one year, nearly two thirds of the patients showed symptom improvement and stabilization of the disease were achieved in around one third of cases; only one subject experienced disease progression. Notably, female gender was a statistically significant factor for improvement, while age and comorbidities did not show statistical significance. A case report from Franic et al. also documented histological improvement after two injections of autologous PRP, with restoration or the physiological dermal and epidermal architecture [31]. On the contrary, an Italian group monitored the effectiveness of PRP therapy in the treatment of vulvar LS with non-invasive methods, proposing video thermography as a suitable tool in this setting [32].

However, it is important to note that the available data on PRP’s effectiveness for genital lichen sclerosus are limited, and larger, well-designed clinical trials are needed to establish its long-term benefits and safety. Moreover, the degree of improvement can vary among individuals, and not all patients may experience the same level of relief. While much of the research on PRP therapy for lichen sclerosus has focused on female patients, there is a growing interest in its applicability for males. Preliminary studies and clinical observations in male patients have suggested that PRP therapy may offer some degree of relief from the symptoms associated with lichen sclerosus. A recent study by Casabona et al. aimed to assess the effectiveness of PRP injections in treating penile LS [25]. Forty-five male patients with penile LS received autologous PRP injections, and various factors including age at diagnosis, treatment frequency, clinical conditions, symptoms, and treatment outcomes were examined. The results demonstrated that PRP treatment led to significant improvements in clinical conditions and symptom reduction in all patients. Topical steroid therapy, which was previously used, was not resumed after PRP treatment. The Investigator’s Global Assessment (IGA) and the Dermatology Life Quality Index (DLQI) both showed substantial improvements after PRP treatment. Another prospective multicentric study evaluated the safety and effectiveness of autologous PRP treatment for LS in male subjects refractory to conventional therapy for at least six months [33]. Five patients met the inclusion criteria and underwent PRP injections every eight weeks. After 18 months, significant decrease in IGA and DLQI was observed, although visual changes were minimal. All patients reported being symptom-free at 10 months, with only one patient experiencing a complication (balanitis). Possible gender-related differences were investigated in a very recent paper from Tedesco and coauthors [34]. A total of 43 males and 51 females received PRP treatment (three infiltrations, 15 days apart) for genital LS. The PRP treatment was well-tolerated, resulting in a significant overall symptom reduction after six months. Pain and burning sensation improved significantly in both genders, with a more pronounced effect in women. Itching reduction was similar in both sexes, while dyspareunia (pain during sexual intercourse) improved significantly only in males. Taken together, these data suggest that PRP therapy may play a crucial role in managing LS by improving quality of life and sexual function for both genders, while also highlighting gender-related differences in symptom severity and age of onset. However, further research, including larger clinical trials specifically focused on male patients, are needed to determine the long-term efficacy and safety of PRP therapy in this setting.

## 3. Adipose-Derived Stem Cell (ADSC) Therapy

Mesenchymal stem cells, nowadays more precisely defined as mesenchymal stromal cells (MSCs), represent a fascinating tool in regenerative medicine for their regenerative and immunomodulatory properties [35]. MSCs are multipotent cells present in tissues of mesenchymal origin [35,36], characterized by the presence of specific surface markers [37,38,39] such as CD105, CD73, and CD90 and by the lack of other lineage-specific markers (CD45, CD34, CD14, CD19, and HLAII). Their multipotency and their regenerative properties are demonstrated by their ability to differentiate into adipocytes, chondrocytes, and osteocytes in vitro. MSCs also interact with lymphocytes, dendritic cells, and macrophages, therefore modulating cytokine secretion (see Figure 2).

MSCs include many different cell types, with probably the most widely studied being the bone-marrow stromal stem cells (BMSCs) [40]. However, MSCs have been found in many other tissues and organs, including muscles, skin, placenta, gingiva, and many others. Among these, adipose-derived stem cells (ADSCs) are of particular interest for their clinical use due to potential advantages over BMSCs as a therapeutic cell source. First of all, the isolation procedure is safer [41,42]. Secondly, the average frequency of MSCs in processed lipoaspirate is much higher than in BM (approximately 2% vs. 1 in 25,000–100,000 nucleated cells) [43]. These characteristics bring the possibility of transplanting freshly isolated ADSCs without the need of cell culturing, therefore not only lowering the risk of microbial contamination, but also preventing cellular alterations and processing errors. ADSCs are currently considered important candidates in regenerative medicine not only because of their plasticity and their regenerative capabilities [36,44,45], but also for directly interacting with immune cells and modulating cytokine secretion [46,47,48,49,50].

ADSC therapy has garnered attention for its potential in the treatment of lichen sclerosus. Preliminary studies have explored the use of adipose-derived stem cells (ADSCs) as a therapeutic approach to address the complex pathophysiology of the condition [16]. The regenerative potential of ADSCs is particularly relevant in the context of lichen sclerosus, where tissue damage, fibrosis, and scarring are common features [51]. Preliminary studies suggest that ADSC therapy may help modulate the immune response, reduce inflammation, and promote tissue healing. This immunomodulatory effect may help alleviate the chronic inflammation that contributes to symptoms such as itching, pain, and discomfort. Moreover, another potential advantage of ADSC therapy is its ability to address the fibrotic changes that occur in LS [52]. Fibrosis, or the formation of scar tissue, is a hallmark of the condition and invariably leads to structural changes which contribute to patient discomfort and soreness [53].

In addition to this, ADSCs do not exploit their regenerative and immunomodulatory function only when isolated and subsequently injected but also when transplanted together with other components of the fat tissue. This happens in the setting of lipofilling, nano- or micro-fat grafting and SVF (stromal vascular fraction) transplantation [54]. These procedures enable clinicians to at least partially restore the normal genital anatomy and potentially lead to better cosmetic outcomes, with positive implications in terms of quality of life and patient satisfaction. In fact, structural changes can lead to functional impairment and aesthetic concerns, and the regenerative properties of ADSCs offer the potential to reverse or mitigate these alterations. This includes the restoration of normal tissue architecture and the reduction of fibrosis, which can be essential for maintaining both physical and psychological well-being [52].

ADSC-based therapies have already been proven to be effective in the setting of chronic vulvar trophism alterations associated with physiological and pathological conditions and the available literature suggests that ADSCs hold promise for treating vulvar dystrophies, primarily due to ADSCs’ angiogenic and trophic properties [55]. A prospective cohort study was conducted on 33 women to assess the impact of a single lipotransfer treatment in women with fibrosis and scarring due to LS [56]. Sexual function, symptoms, pain, psychological status, and quality of life were evaluated before and after treatment. The results showed significant improvement in all the aforementioned parameters according to patient judgment, and these findings were confirmed by physician-based assessment, which indicated overall improvements in the treated areas.

Boero and collaborators published a prospective study on 36 patients with LS who were offered fat grafting [57]. All of them had not responded to previous treatments. The primary outcomes included evaluating improvements in mucocutaneous trophism, symptoms resolution/reduction, and histological changes in the vulvar skin after treatment. Secondary outcomes focused on improvements in quality of life and sexual function. The results showed that 94% of patients experienced improved vulvar skin and mucosal trophism, 75% had enhanced vaginal introitus caliber and elasticity, 50% had reduced clitoris burying, 83% reported increased labia majora and minora volume, 94% had complete disappearance of scratching lesions, and 78% showed remission of white lesions. Additionally, 95% of patients no longer required routine use of topical corticosteroids. Significant improvements in terms pf DLQI and FSFI were detected in all cases. The same group also recently confirmed the effectiveness of fat grafting for the treatment of genital LS longer term [58].

Retrospective evaluation of 39 patients affected by vulvar LS treated with autologous fat grafting enriched with adipose-derived stromal vascular fraction (SVF) indicated that 94.87% of patients experienced a significant decrease in their global scores at both six and 24 months after treatment (*p* < 0.05) [59]. A modified vulvo-vaginal symptoms questionnaire was used to assess symptoms, signs, and functional impairment before treatment, six months after treatment, and 24 months after treatment. This retrospective study demonstrated that autologous fat grafting enriched with adipose-derived SVF is a safe and effective treatment that leads to significant and long-lasting improvements in patients with vulvar lichen sclerosus.

While the preliminary findings regarding ADSC therapy for lichen sclerosus are promising, it is essential to acknowledge that the research is still in its early stages. The existing studies are relatively small in scale and often lack long-term follow-up data. Therefore, although ADSC therapy holds potential as a treatment option, further investigation through well-designed clinical trials is necessary to establish its efficacy, safety, and optimal protocols for addressing the multifaceted challenges posed by lichen sclerosus.

## 4. Combined Treatment: ADSCs and PRP

The rationale for the combined use of PRP and ADSCs for LS resides in previous reports of a possible synergistic action of these regenerative strategies in other settings (such as wound healing or orthopedics) [60,61]. Lipofilling-with the additional injection of PRP has already been proposed as a successful technique to treat vaginal atrophy, which is often associated with LS [62,63]. The first description of combined use of PRP and ADSCs for the treatment of LS dates back to 2010 [64]. Fifteen female patients with LS, previously unresponsive to steroid therapy, were treated. A blood sample was taken, and PRP was obtained through centrifugation. Liposuction was performed from a donor area, and the collected fat was processed and injected into the affected area. PRP was also injected into the same regions. Follow-up ranged from 6 to 24 months. No adverse events occurred, and patients experienced moderate pain for about ten days after the procedure. Within 15 days after the intervention, symptoms such as itching and burning improved, with vulvar skin and mucosa becoming more elastic and normal in appearance. After four months, all patients reported the complete disappearance of pain and symptoms, with normal anatomical features of the vulva. Sexual activity was fully regained. As a side note, patients with more severe fibrosis and atrophy underwent the procedure one or two more times, with satisfactory and stable results.

One of the largest casuistries of patients affected by genital LS treated with combination of PRP and fat grafting has recently been published by Casabona et al. [9]. The presented study aimed to assess the impact of a combined treatment approach on the quality of life of patients affected by vulvar lichen LS. The clinical records of 72 subjects who underwent regenerative surgery were reviewed. Various quality of life measures were used, including the Dermatology Life Quality Index (DLQI), Skindex-29, Female Sexual Function Index (FSFI), and patient-administered clinical scoring system (CSS). All assessment scores significantly improved, demonstrating a positive impact on patients’ quality of life.

In a milestone prospective pilot trial by Gutierrez-Ontalvilla, 20 patients diagnosed with moderate to severe vulvar LS were divided into two groups [65]. The treatment group (TG) received two injections of nanofat mixed with platelet-rich plasma (PRP) into the vulvar area at three-month intervals, while the control group (CG) received standard topical corticosteroid therapy. Fat was obtained from the thigh or abdomen and processed into nanofat suspension. This study assessed treatment efficacy by measuring changes in vulvar skin elasticity, histopathology, clinical symptoms, signs, and patient quality of life after one year. Only 19 patients completed the study protocol (9 TG and 10 CG). After one year, the TG demonstrated significant improvement in terms of symptoms (itching, pain, burning, and dyspareunia) and clinical signs (cervical erosions, fissures, stenosis, and leukoderma). Biopsy analysis revealed a significant decrease in the inflammatory infiltrate in the TG. However, there was no significant improvement in vulvar skin elasticity. Notably, no adverse events related to the autologous treatment were reported. In conclusion, compared to topical corticosteroids, the autologous treatment reduced vulvar inflammation and improved most clinical symptoms and signs associated with VLS, although vulvar skin elasticity did not improve.

In a study published in 2020, the authors compared the efficacy of adipose-tissue derived stromal vascular fraction (AD-SVF) and AD-SVF enriched with PRP, in the setting of genital lichen sclerosus (LS) in 40 symptomatic patients who were unresponsive to medical treatment [66]. Patients, aged 43–78 years, were randomly assigned to two groups and evaluated using the DLQI at baseline and six months after treatment. Both therapeutic approaches showed a strong safety profile with no complications and determined significant clinical improvement at a six month follow-up. However, combinatory therapy was less effective in late-stage patients. The authors concluded that combinatory therapy could be recommended for early-stage patients due to a synergistic effect in this specific context but discouraged the use of AD-SVF plus PRP for late-stage patients.

## 5. Conclusions

Fat grafting appears to be a valuable supplementary treatment for selected cases of vulvar LS, especially in patients who do not respond to first-line therapies or have severe anatomical impairments affecting sexual function and quality of life. The effectiveness of lipofilling and other fat grafting techniques at least partially resides in the regenerative potential of ADSCs [52]. PRP therapy also holds promise as a potential treatment for lichen sclerosus, with preliminary evidence suggesting its efficacy in symptom relief and potential structural improvement. Its safety profile and regenerative properties make it an attractive option for individuals seeking alternatives to conventional therapies.

The effectiveness of PRP and ADSC based therapies in the management of genital lichen sclerosus is a topic of growing interest within the field of dermatology and regenerative medicine [16]. Current evidence does not enable us to indicate whether one technique is preferable to the other in specific clinical contexts, such as in the presence of comorbidities. Both therapies are relatively new, and more extensive research is needed to determine their long-term effectiveness, safety, and optimal protocols for treating this complex and chronic skin condition.

Despite the promising results, the present paper intrinsically brings to our attention some limitations of the ongoing research in the setting of regenerative strategies for LS. The high heterogeneity of the published casuistries, the lack of common endpoints, and shared protocols among the available studies make it difficult to draw consistent conclusions, which gives reason to the present review for being narrative end not enabling further metanalysis. Moreover, most of the available literature on this topic is composed of short reports and research letters which do not contain a detailed description of the methodology used, making it impossible to provide complete and comparable data in terms of outcome and FUP among published casuistries. The available studies are relatively small in scale, and the duration of follow-up is often limited. The duration of the follow-up, for example, available in just over half of cases, ranged from 1 up to 24 months. Therefore, while the initial results are promising, more extensive and well-designed clinical trials are necessary to establish the long-term efficacy and safety of PRP therapy in the management of lichen sclerosus. Study endpoints also bring a significant bias in the interpretation of published results, as only few studies use validated scales to assess clinical efficacy; among these, further analysis is limited by the presence of high variability in chosen parameters. However, CSS, DLQI, and FSFI should be considered currently for wider use in clinical research (see Appendix A)

Potential limitations of the use of ADSCs and PRP reside in the real-life feasibility of such treatments in a broader clinical setting. Patients considering these therapies should consult with qualified dermatologists or healthcare providers who can provide guidance based on the most current scientific evidence and clinical experience. Additionally, it is essential to consider individual factors, including the severity of the condition and any underlying health issues, when exploring these innovative treatment options.

## 6. Future Directions

Despite the promising results achieved in the setting of LS, the use of PRP and ADSCs on a large scale remains far from being close to happen in the clinical daily practice. Shared protocols and official guidelines are urgently needed. Future research with multicenter international randomized controlled trials will likely bring new evidence in order to better define which LS cases should be treated with PRP and/or ADSCs, and the protocols for their administration.

## Figures and Tables

**Figure 1 ijms-24-16107-f001:**
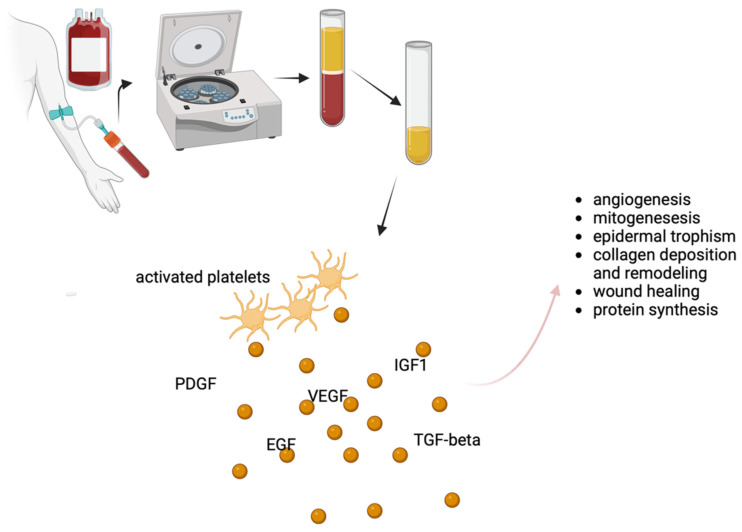
Schematic representation of the use of PRP in the setting of LS and possible mechanisms of action. PRP is obtained through centrifugation of blood and its content is particularly rich in platelets. When injected, activated platelets release a series of growth factors responsible for their therapeutic efficacy. PDGF: platelet-derived growth factor; EGF: epidermal growth factor; VEGF: vascular endothelial growth factor; IGF: insulin-like growth factor; TGF: transforming growth factor. Created with BioRender.com, accessed on 14 October 2023.

**Figure 2 ijms-24-16107-f002:**
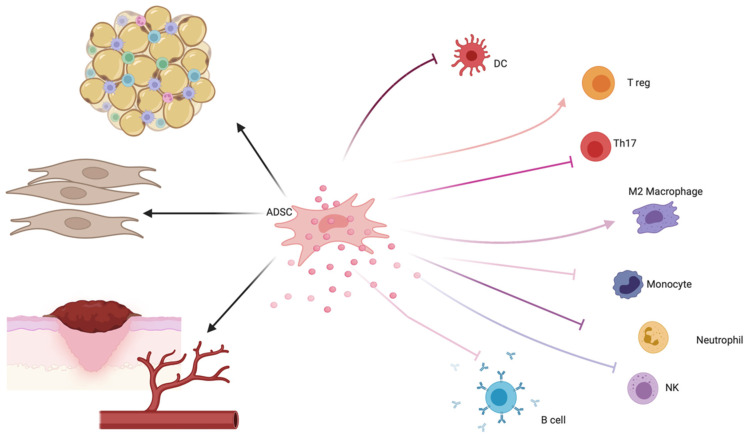
Possible mechanisms of action of ADSCs in the setting of LS. ADSCs exert an immunomodulatory action through the secretion of anti-inflammatory cytokines and have regenerative functions. In fact, ADSCs can both differentiate towards a fibroblast-like phenotype and activate local fibroblasts. At the same time, ADSCs contained in the adipose tissue can differentiate into adipocytes, which is fundamental for restoring volumes in the genital area. Finally, ADSCs are implied in tissue healing promoting both epithelization and vessel formation. Created with BioRender.com, accessed on 14 October 2023.

**Table 1 ijms-24-16107-t001:** Main publications retrieved with our search. The name of the first author is indicated, as well as the year of publication. The number of described patients is also indicated according to sex and type of treatment received. In total, data on 526 patients were available. Y: year; M: males; F: females; ADSCs: adipose tissue-derived stromal cells; PRP: platelet-rich plasma.

Author	Y	M	F	ADSCs/Fat	PRP	ADSC + PRP
Almadori	2020		33	33		
Behnia-Willison	2016		28		28	
Boero	2015		36	36		
Casabona	2010		15			15
Casabona	2017	45			45	
Casabona	2023		72			72
Cohen	2019		1			1
Franic	2018		1		1	
Gutierrez-Ontalvilla	2022		9			9
Goldstein	2016		15		15	
Goldstein	2019		19		19	
Kim	2017		1			1
Medina Garrido	2023		28		28	
Monreal	2020		39	39		
Navarrete	2020	5			5	
Onesti	2016		8	8		
Tedesco	2019	13	18		31	
Tedesco	2020		40	20		20
Tedesco	2021		6		6	
Tedesco	2022	43	51		94	
		106	420	136	272	118

## Data Availability

Data are available from the authors upon reasonable request.

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
