# Peer review of "Platelet-Rich Plasma (PRP) and Adipose-Derived Stem Cell (ADSC) Therapy in the Treatment of Genital Lichen Sclerosus: A Comprehensive Review"

_ijms, 2023, doi:10.3390/ijms242216107_

Round 1
Reviewer 1 Report
Comments and Suggestions for Authors
The manuscript provides a concise review about the novel and relatively scarcely used treatment modalities for genital lichen sclerosus et atrophicus (LS) - platelet plasme injections (PRP) and adipose-derived stromal cells (ADSC). Such reviews are to date scarce in the published literature and to the knowledge of this reviewer such comparative analysis of both of these treatment modalities does not exist. The methodolgy used was sound, the literature analysis was performed in a rigorous manner and the conclusions drawn are well in coherence with the data presented. As the literature regarding the use of either PRP or ADSC is sitll relatively modest the references analysed are near-exhausting. The manuscript includes two schematic figures that illustrate the basic mechanisms by which PRP and ADSC would contribute to the healing of LS. The Images are appropriate and correspond well to the styoe of this review.
Author Response
We thank the Reviewer for his/her kind comments.
Reviewer 2 Report
Comments and Suggestions for Authors
In this manuscript, Alessia Paganelli and colleagues explore and summarize the utilization of PRP and ADSC therapies in the treatment of genital LS, including their mechanisms of action, safety profiles, and clinical outcomes. I found this review is interesting, but there are some critical issues that the authors should address. I detail my criticisms below:
1. For Table 1: since this is a Comprehensive Review, authors should list the outcomes of the published studies and how long the studies have been followed up.
2. It would be better for increasing the readability if authors apply a table to list the different between Autologous and homologous PRP.
3. Paragraph layout and logic issues in manuscript: The authors list too many paragraphs, and they should be integrated based on logic.
1) Introduction part, paragraphs 1-3 and 5 are mainly talk about the basic information and symptoms of Lichen sclerosus, they should be integrated as paragraph 1.
2) Introduction part, paragraphs 4, 6 and 7-9 are talking about the etiology and current treatment for lichen sclerosus, they should be integrated as paragraph 2.
3) Introduction part, paragraphs 10 as a separate paragraph.
4) Authors should do the same modifications for the remaining content at lines 122-365.
4. Authors should summarize the advantages and limitations for PRP and ADSC therapies to treat Lichen sclerosus in Conclusion part.
Author Response
- We thank the reviewer for his/her kind suggestion. Unfortunately specific study outcomes and FUP duration are not always included in the selected studies. The present review aims at being descriptive rather than systematic, which would be very hard due to high heterogeneity of published casuistries. In fact, many of these papers are only letters to the editor and do not contain a detailed description of used methodology and therefore make it impossible to provide comparable data among published casuistries in terms of outcome and FUP. We added a supplementary table (Suppl file 2) in support of these considerations. We also added few lines to the conclusions clarifying this point, in order to help the reader in the interpretation of the presented data.
-
To our knowledge, no clinical studies have been published so far on the use of homologous PRP in the setting of LS. We clarified this point in the text.
-
Thank you very much for these valuable suggestions. We simplified paragraph disposition throughout the text.
-
Thank you very much for your suggestion, we added some considerations in the conclusions.
Reviewer 3 Report
Comments and Suggestions for Authors
In this review, Paganelli et al reported summarized current evidence of Platelet-Rich Plasma (PRP) and Adipose-Derived Stem Cells (ADSC) therapy as potential regenerative therapies for lichen sclerosus (LS). This is a very interesting experiment, but there are a few points that are unclear, and my questions are as follows.
major concerns)
1) This is a narrative review. The search strategy and the keywords used in the search were unclear. If the number of searches and the subsequent flow of searches could be summarized using PRISMA, it would be possible to create a more robust systematic review rather than a narrative review. A network meta-analysis may also be possible by summarizing the results of clinical trials, etc.
minor concerns)
1) In line 152, you wrote "twoß years 28 patients with confirmed vulvar LS were treated with autologous PRP[26]. All the patients were non-responders to topical..." What do you mean twoß? Is there a typo? Please check and correct it appropriately.
Author Response
1. Thank you very much for this comment, which enabled us to perform some key changes in the text to better clarify both the aim of the paper and the search strategy. We better specified the search strategy for the papers contained in Table 1 (which is not totally exhaustive of the treated topics, but just summarizes the number of treated patients stratified according to sex and type of treatment). The present review aims at being descriptive rather than systematic, given the high heterogeneity of published casuistries. In fact, many of these papers are only letters to the editor and do not contain a detailed description of used methodology and therefore make it impossible to provide comparable data among published casuistries in terms of outcome and FUP. We added a supplementary file containing PRISMA flow chart (Suppl. File 1) and a detailed table (Suppl file 2) in support of these considerations. We also added few lines to the conclusions clarifying this point, in order to help the reader in the interpretation of the presented data. 2. We checked it and corrected. It was a typo error. Thank you again for the comment.Round 2
Reviewer 2 Report
Comments and Suggestions for Authors
My comments have been siginificantly addressed.